# A Simple Strategy Stabilizing for a CuFe/SiO₂ Catalyst and Boosting Higher Alcohols' Synthesis from Syngas

Nana Gong [1,2], Yingquan Wu [1,*], Qingxiang Ma [3] and Yisheng Tan [1,*]

1 State Key Laboratory of Coal Conversion, Institute of Coal Chemistry, Chinese Academy of Sciences, Taiyuan 030001, China
2 University of Chinese Academy of Sciences, Beijing 100049, China
3 State Key Laboratory of High-Efficiency Coal Utilization and Green Chemical Engineering, Ningxia University, Yinchuan 750021, China
* Correspondence: wuyq@sxicc.ac.cn (Y.W.); tan@sxicc.ac.cn (Y.T.); Tel.: +86-351-404-4287 (Y.T.)

**Abstract:** Stable F-T-based catalyst development in direct CO hydrogenation to higher alcohols is still a challenge at present. In this study, CuFe/SiO₂ catalysts with a SiO₂ support treated with a piranha solution were prepared and evaluated in a long-term reaction. The treated catalyst showed higher total alcohols' selectivity and great stability during a reaction of more than 90 h. It was found that the treatment with the piranha solution enriched the surface hydroxyl groups on SiO₂, so that the Cu–Fe active components could be firmly anchored and highly dispersed on the support, resulting in stable catalytic performance. Furthermore, the in situ DRIFTS revealed that the adsorption strength of CO on Cu⁺ on the treated catalyst surface was weakened, which made the C-O bond less likely to be cleaved and thus significantly inhibited the formation of hydrocarbon products. Meanwhile, the non-dissociated CO species were obviously enriched on the Cu⁰ surface, promoting the formation of alcohol products, and thus the selectivity of total alcohols was increased. This strategy will shed light on the design of supported catalysts with stabilized structures for a wide range of catalytic reactions.

**Keywords:** higher alcohols; syngas; CO hydrogenation; CuFe/SiO₂; stability; hydroxyl group; piranha solution

## 1. Introduction

The direct combustion of coal leads to serious environmental pollution, and the synthesis of higher alcohols (HAS) from coal by synthesis gas (H₂ + CO) is one of the ways for the efficient and clean utilization of coal, which has considerable application prospects. Higher alcohols can not only be used as gasoline additives or gasoline substitutes, but can also be separated to obtain chemicals with high practical value, such as ethanol, propanol, and butanol. The synthesis of higher alcohols (HAS) from syngas was first reported in the 1920s [1]. So far, researchers have developed several types of catalyst systems, including (1) modified methanol catalysts (Zn-Cr and Cu-based catalysts) [2–6], (2) modified Fisher–Tropsch (F-T) catalysts [7–11], (3) noble metal catalysts (Rh) catalysts [12–14], and (4) Mo-based catalysts [15–20].

Among the above catalyst systems, Cu–Fe-based catalysts, as one kind of modified F-T catalysts, have attracted considerable attention due to their low price, excellent activity, and good selectivity for total alcohols, especially higher alcohols under mild conditions [21–26]. It is generally accepted that bi-functional active sites are required for higher alcohol synthesis; one is metallic Cu, for the non-dissociative adsorption of CO, and the other is Fe, for the dissociative adsorption of CO to form CHₓ species. The higher alcohols can be formed by CO insertion into CHₓ species and successive hydrogenation [21,27]. As can be seen, achieving the intimate combination of Cu and Fe species to promote their synergetic effect is the key to efficient higher alcohol synthesis from CO hydrogenation. However, the long-term stability of Cu-Fe catalysts is commonly poor, and the deactivation of the catalyst

will lead to a decrease in total alcohol selectivity, which is very challenging for further work [28,29]. At the same time, some studies further studied the deactivation mechanism and revealed that the deactivation of Cu-Fe catalysts was attributed to the separation of the Cu phase and Fe phase, which weakened the synergy between Cu and Fe, thus resulting in a decrease in the alcohols' selectivity [28].

The supported catalysts prepared by an impregnation method are not only simple to obtain, but they also have high utilization rates and high dispersion of the active components on the surface of the carrier. Despite all of these, the active components are prone to agglomerate during the preparation, reduction, and reaction process due to the weak interaction between active components and the carrier, which inhibited the catalytic activity and stability. $SiO_2$ is usually selected as a catalyst carrier owing to its large specific surface area for highly dispersing active components and its inert property for clearly revealing the interaction between active phases [30–33]. It should be noted that the surface properties of the carrier have obvious effects on the dispersion of active components, such as surface hydroxyl groups, which play an important role in the dispersion and anchoring of active components, and will then affect the catalytic performance and stability of the catalyst [34–36].

In this study, we report an effective strategy to enhance the stability of the $CuFe/SiO_2$ catalyst and boost the synthesis of higher alcohols from syngas by pretreating $SiO_2$ with piranha solution. The effect of the surface properties of the $SiO_2$ carrier on the dispersion of active components and the synthesis performance of higher alcohols before and after the treatment were investigated.

## 2. Results and Discussion

### 2.1. Textural Properties of the Catalysts

Table 1 summarizes the results from $N_2$ physisorption for the prepared catalysts. It should be noted that the surface area of $SiO_2$ exhibited a significant increase after being treated with piranha solution. In addition, the average pore size was slightly reduced, indicating that new, small pores may be formed in $SiO_2$ after piranha solution treatment. When the $SiO_2$ was loaded with CuFe, the specific surface areas were all significantly reduced, indicating that some CuFe entered in the $SiO_2$ pore channels and blocked some of them, resulting in a decrease in pore volume.

**Table 1.** Detailed physicochemical properties of the calcined catalysts.

| Catalyst | BET Surface Area ($m^2/g$) | Pore Volume ($cm^3/g$) | Average Pore Diameter (nm) |
|---|---|---|---|
| $SiO_2$-N | 522 | 0.70 | 5.40 |
| $SiO_2$-T | 620 | 0.79 | 5.08 |
| $CuFe/SiO_2$-N | 413 | 0.54 | 5.19 |
| $CuFe/SiO_2$-T | 434 | 0.55 | 5.11 |

### 2.2. Surface Groups on $SiO_2$ before and after Treatment

Fourier transform infrared (FTIR) spectroscopy measurements were carried out on a Tensor 27 infrared spectrometer (Bruker, Bremen, Germany) to examine the surface groups including hydroxyl groups (-OH), organic groups, and silicon–oxygen bonds (Si-O) on $SiO_2$ before and after piranha solution treatment (Figure 1).

As is shown, the broad bands from 3000 $cm^{-1}$ to 4000 $cm^{-1}$ are assigned to the vibrations of –OH groups. The characteristic peak above 3700 $cm^{-1}$ is attributed to the terminal hydroxyl groups, while a broad peak below 3700 $cm^{-1}$ is muti-coordinated hydroxyl groups [34,37–40]. It can be seen from the figure that before treatment, there were more terminal hydroxyl groups on the surface of $SiO_2$-N. After piranha solution treatment, the characteristic peak intensity above 3700 $cm^{-1}$ decreased significantly, while the broad peak intensity increased, indicating that the multidentate hydroxyl groups on the surface

of $SiO_2$ increased, resulting in the red shift of the absorption peak. In addition, it should be noted that several obvious infrared absorption peaks in the range of 1000–2000 cm$^{-1}$ located at 1862, 1624, 1356, and 1295 cm$^{-1}$ were ascribed to the stretching vibration of organic groups in $SiO_2$. The bands at 1197 and 1045 cm$^{-1}$ are characteristic of the stretching vibrations of the Si–O bond. It can be found that the strength of peaks for the Si–O bond changed little, while the characteristic peaks of the organic groups decreased obviously after treatment. This indicated that some unknown organic groups on the surface were removed after being treated with piranha solution, giving rise to the larger surface area, as the BET results showed, which may provide new sites for the formation of multi-coordinated -OH on the surface of $SiO_2$.

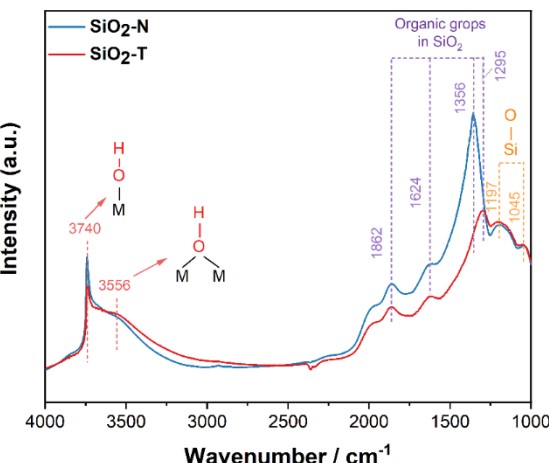

**Figure 1.** FTIR spectra of surface species on $SiO_2$ before and after piranha solution treatment.

Hence, one can see that after piranha solution treatment on $SiO_2$, on the one hand, the specific surface area was obviously increased, and on the other hand, some unknown groups on the surface were removed and more multi-coordinated hydroxyl groups were produced, which might be beneficial to the subsequent metal dispersion and anchoring on the surface of $SiO_2$.

### 2.3. Crystalline Phase of the Catalysts

From the XRD patterns of the fresh catalysts (Figure 2a), it can be found that all of the fresh catalysts showed diffraction peaks at 2θ of 35.6°, 38.8°, 48.7°, and 33.4°, corresponding to crystal CuO (JCPDS card No. 45-0937) and α-$Fe_2O_3$ (JCPDS card No. 33-0664), respectively [41–43]. It should be noted that $CuFe_2O_4$ is usually formed after calcination and the diffraction peak of $CuFe_2O_4$ (JCPDS card No. 34–0425) located at 35.6° may overlap with CuO [22]. By comparing the CuFe/$SiO_2$-N and CuFe/$SiO_2$-T catalysts, the structure of $SiO_2$ was almost unchanged after the treatment, but the diffraction peaks attributed to CuO and $Fe_2O_3$ on CuFe/$SiO_2$-T were slightly weakened, indicating that Cu and Fe were better dispersed on $SiO_2$-T and the catalyst particle size became smaller after treatment. Moreover, the XRD patterns of the used catalyst are shown in Figure 2b, from which CuO was reduced to Cu$^0$ after the reaction, and $Fe_2O_3$ seemed to be reduced to low valence compounds. The diffraction peaks of the untreated CuFe/$SiO_2$-N after the reaction were sharp and intense, indicating the agglomeration or crystallization of Fe species. It can be concluded that whether the catalysts were fresh or used, Cu and Fe species were better dispersed on the surface of CuFe/$SiO_2$-T than on CuFe/$SiO_2$-N, suggesting that the active components can be better dispersed and anchored on the surface of $SiO_2$ after piranha solution treatment.

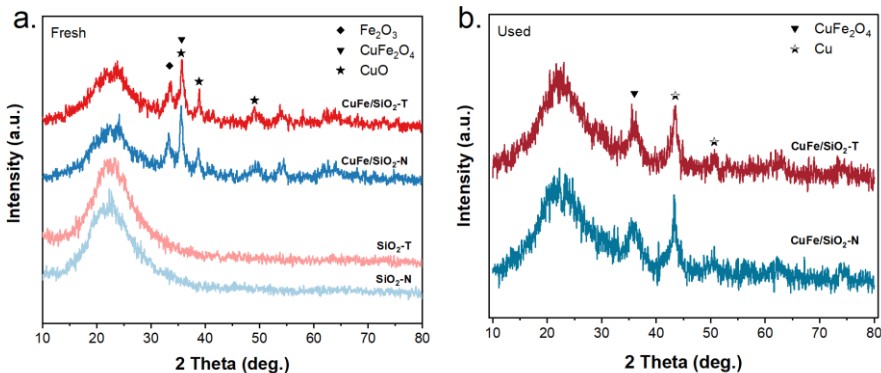

**Figure 2.** XRD patterns of the CuFe/SiO$_2$ samples: (**a**) fresh samples and (**b**) used samples.

### 2.4. Surface Chemical Environment of the Catalysts

The chemical environment of Cu and Fe species was studied by XPS. From the XPS spectra for the Cu 2p of the calcined catalysts (Figure 3a), two peaks at ~932.99 eV and ~952.78 eV attributed to the spin-obit doublet of Cu 2p were observed, which can be assigned to the binding energies of Cu 2p$_{3/2}$ and Cu 2p$_{1/2}$ in CuO, respectively [39]. The other two peaks on the higher binding energy side of both Cu 2p$_{3/2}$ and Cu 2p$_{1/2}$ are satellite structures. From the XPS spectra for the Fe 2p of the calcined catalysts (Figure 3b), the binding energies of Fe 2p$_{3/2}$ and Fe 2p$_{1/2}$ are at ~710 eV and ~723 eV, respectively, corresponding to Fe$_2$O$_3$ [24]. Based on these results, it can be concluded that the copper and iron species in all of the calcined catalysts mainly existed as Cu$^{2+}$ and Fe$^{3+}$, respectively, which is in accordance with the XRD results.

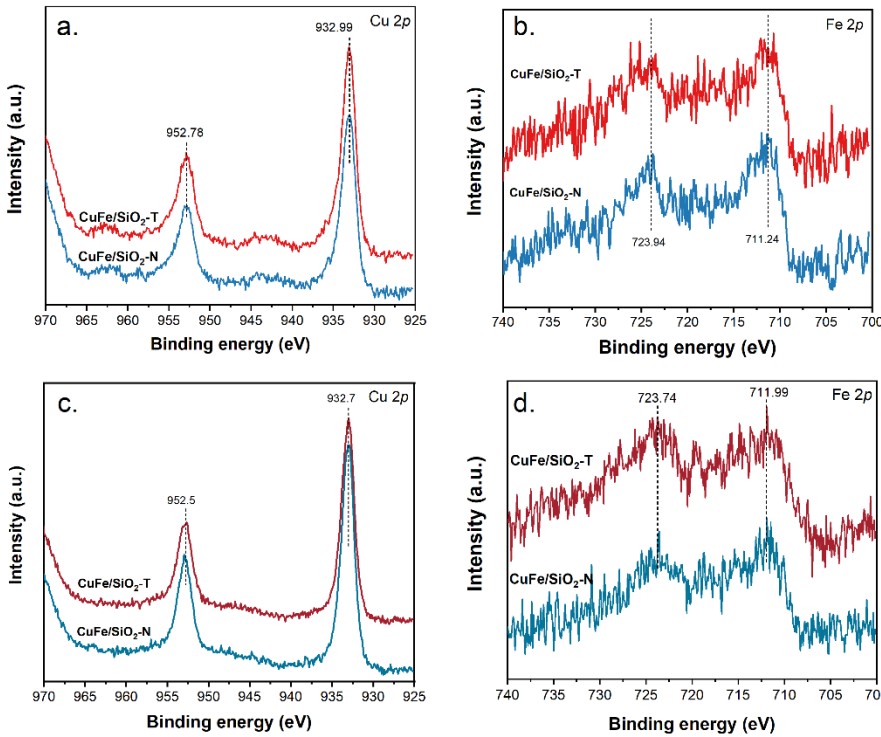

**Figure 3.** XPS spectra of the fresh and reduced catalysts: (**a**) Cu 2p spectra of fresh samples; (**b**) Fe 2p spectra of fresh samples; (**c**) Cu 2p spectra of used samples; and (**d**) Fe 2p spectra of used samples.

The XPS spectra for the Cu 2p and Fe 2p of the used catalysts are shown in Figure 3. The binding energies of Cu 2p$_{3/2}$ and Cu 2p$_{1/2}$ were located at 932.7 eV and 952.5 eV, respectively, which can be attributed to the characteristic of Cu$^0$ (Figure 3c) [44]. The disappearance of the shake-up satellites of Cu$^{2+}$ species verifies that Cu$^{2+}$ species were

completely reduced to $Cu^0$ species after the reduction process, which is in agreement with the results of XRD (Figure 2). On the other hand, the change of Fe 2p was not obvious before and after the reaction (Figure 3d), which was also consistent with XRD, indicating that $Fe_2O_3$ was almost not reduced during the reduction and the reaction process. Of note, no obvious shift of peak position was found in all of the samples. This suggested that the interaction effect of Cu and Fe species was not significantly affected after the treatment with piranha solution.

Furthermore, the surface elemental distribution was also calculated from XPS patterns. In our preparation process, the catalysts were all prepared with a Cu/Fe ratio of 1:1. In other words, if no agglomeration occurs on the catalyst's surface, the elemental ratio of Cu and Fe should theoretically be 1:1. As shown in Table 2, the traditionally prepared $CuFe/SiO_2$-N catalyst showed the Cu/Fe molar ratio of 1.16 (>1), indicating that a Cu component accumulated on the catalyst surface after the preparation and calcination process. In contrast, the Cu/Fe molar ratio of the $CuFe/SiO_2$-T catalyst was 0.98, which is close to the calculated value. This implied that the treatment of piranha solution has effectively promoted the dispersion of active components and avoided the enrichment of Cu species on the surface, which may further improve the catalytic stability. These phenomena are consistent with the XRD results.

**Table 2.** XPS results of $CuFe/SiO_2$-N and $CuFe/SiO_2$-T catalysts.

| Catalyst | Relative Surface Concentration of Catalysts (at. %) | | | | Cu/Fe Molar Ratio |
|---|---|---|---|---|---|
| | Cu | Fe | Si | O | |
| $CuFe/SiO_2$-N | 0.88 | 0.76 | 69.26 | 29.10 | 1.16 |
| $CuFe/SiO_2$-T | 0.81 | 0.83 | 68.49 | 29.87 | 0.98 |

### 2.5. The Reducibility of the Catalysts

To reveal the reducibility of the $CuFe/SiO_2$-N and $CuFe/SiO_2$-T catalysts, $H_2$-TPR measurement was carried out and the profiles are shown in Figure 4. A main peak centered below 300 °C can be attributed to the reduction of CuO to metallic Cu [45,46]. After piranha solution treatment, the reduction peak of CuO shifted to a lower temperature from 241 °C to 236 °C, indicating higher dispersed and smaller CuO nanoparticles in the $CuFe/SiO_2$-T catalyst, and almost no obvious reduction peak of $Fe_2O_3$ was observed, which is in consistent with the XRD and XPS results. Moreover, the consumption of $H_2$ on the two catalysts was nearly the same, indicating the same composition of the two catalysts. This further implied that the $CuFe/SiO_2$-T catalyst had better dispersed CuO species on the surface of $SiO_2$.

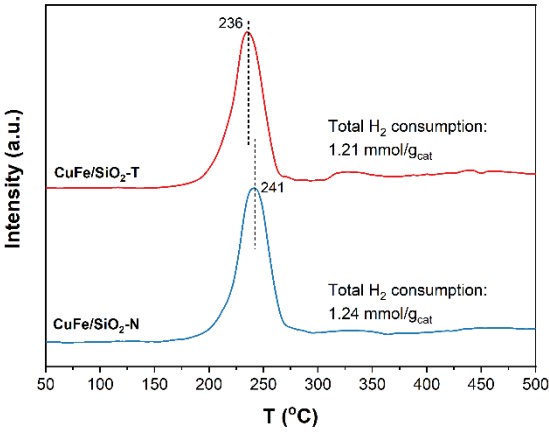

**Figure 4.** $H_2$-TPR profiles of $CuFe/SiO_2$ catalysts before and after treatment.

### 2.6. STEM-EDS Analysis

In order to further understand the effect of piranha solution treatment on the morphology and dispersion of Cu and Fe components, the two samples (CuFe/SiO$_2$-N and CuFe/SiO$_2$-T) were characterized by scanning transmission electron microscopy (STEM) and energy-dispersive X-ray spectroscopy (EDS). It can be seen from Figure 5 that different particle size distributions can be observed on the surface of CuFe/SiO$_2$-N. In comparison, no well-defined particles could be observed on the surface of the CuFe/SiO$_2$-T catalyst, indicating that after treatment of piranha solution on SiO$_2$, Cu and Fe can be more evenly distributed on the surface of the catalyst and would not aggregate to form larger particles.

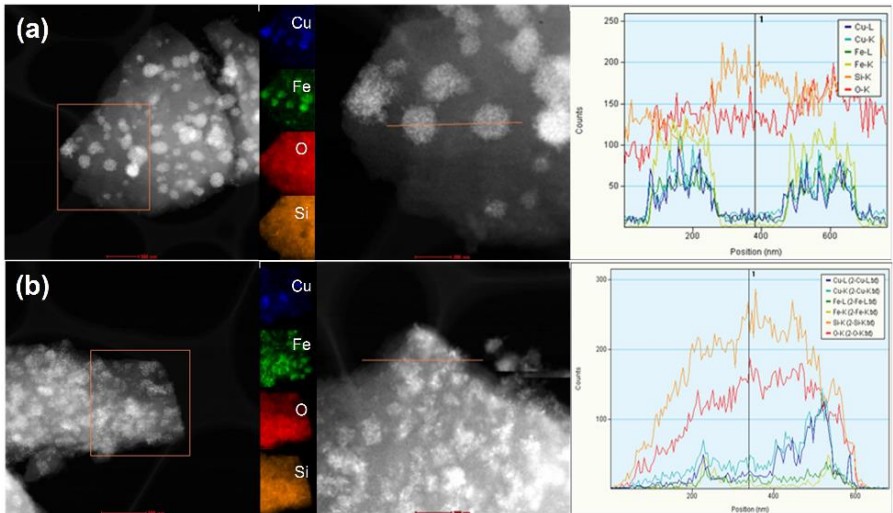

**Figure 5.** HADDF-STEM and elemental mapping images of the CuFe/SiO$_2$ samples: (**a**) CuFe/SiO$_2$-N and (**b**) CuFe/SiO$_2$-T.

### 2.7. In Situ DRIFTS of CO Adsorption

The catalytic performance of the catalyst usually depends on its ability to adsorb and activate CO and H$_2$. Therefore, we investigated CO adsorption by in situ DRIFTS on the CuFe/SiO$_2$-N and CuFe/SiO$_2$-T catalysts.

Figure 6 shows the spectra obtained after CO adsorption on CuFe/SiO$_2$-N and CuFe/SiO$_2$-T at 30 °C. The catalyst was first exposed to a CO atmosphere at a total pressure of 0.1 MPa for 30 min. Then, purge with Ar flow and record the spectra at the same time. It can be found that before Ar purging, the spectrum is mainly composed of two absorption bands at 2000–2300 cm$^{-1}$, corresponding to the gaseous CO species. After switching to Ar purging, the CO species chemisorbed on the catalyst surface remained. It can be seen from CuFe/SiO$_2$-N that when the gas phase CO was purged (after 13 min), an obvious characteristic peak of CO can still be observed at 2121 cm$^{-1}$, which was ascribed to the linearly adsorbed CO on the surface of Cu$^+$ species [47]. This indicated that there was a small amount of Cu$^+$ in the catalyst. Differently, when the catalyst CuFe/SiO$_2$-T was purged by Ar for 11 min, only weak CO adsorption could be observed at 2126 cm$^{-1}$. By comparing the position and intensity of the characteristic peaks of CO after the same purge time, it was found that the adsorption of CO on the surface of the catalyst treated with piranha aqueous solution was obviously weakened. Larger particles can donate more electron density and the resulting CO stretching vibration will shift to a lower frequency [47], thus CO adsorption on CuFe/SiO$_2$-N was stronger than CuFe/SiO$_2$-T. In addition, after piranha solution treatment, there was an obvious shoulder peak at 2089 cm$^{-1}$, which belonged to CO adsorption on Cu$^0$, indicating that there were more Cu$^0$ species on the surface of the CuFe/SiO$_2$-T catalyst, and the content of non-dissociated adsorbed CO on the surface increased significantly. According to the model of CO adsorption over metal surfaces by the Blyholder model [48], when the adsorption of CO was weakened on the catalyst's surface,

the C–O bond was not conducive to fracture, and CO was easy to participate in the reaction in the form of a non-dissociation state, which promoted CO insertion reaction.

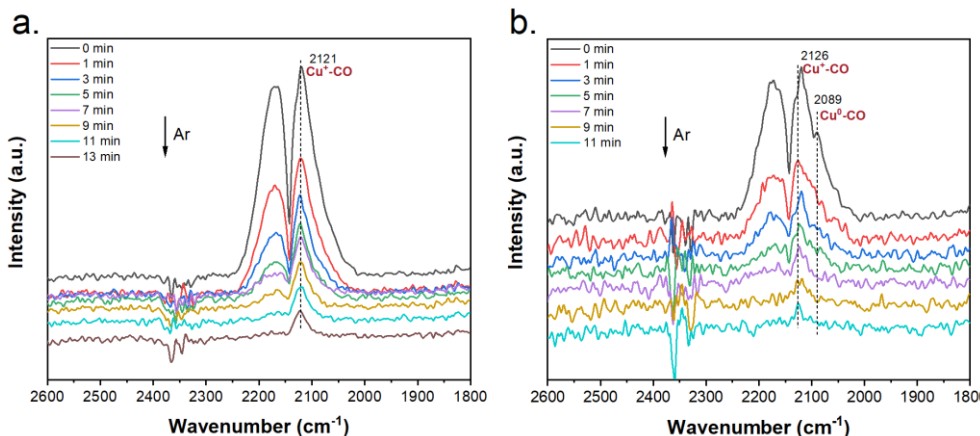

**Figure 6.** In situ DRIFTS of CO adsorption on the CuFe/SiO$_2$ catalysts after reduction: (**a**) CuFe/SiO$_2$-N and (**b**) CuFe/SiO$_2$-T.

### 2.8. CO Hydrogenation over CuFe/SiO$_2$-N and CuFe/SiO$_2$-T Catalysts

The catalytic performance of the CuFe/SiO$_2$-N and CuFe/SiO$_2$-T catalysts for CO hydrogenation in the long-term reaction was evaluated at 250 °C, 5 MPa, and 3000 h$^{-1}$ with H$_2$/CO (H$_2$:CO = 2.5:1, volume ratio) mixed gas. The carbon balance data were calculated 100 ± 3% in the whole reaction process to ensure the validity of the data. The evaluation results are presented in Figure 7.

It can be seen from Figure 7a that with the extension of the reaction time, the CO conversion on the CuFe/SiO$_2$-N catalyst showed a gradually increasing trend, while the CuFe/SiO$_2$-T catalyst showed a steady CO conversion rate of around 40%, indicating the catalytic activity was kept unchanged after piranha solution treatment in the CuFe/SiO$_2$-T catalyst. Figure 7b showed the product selectivity on the CuFe/SiO$_2$-N catalyst. It can be seen that the selectivity of hydrocarbon products (CH$_x$) increased after a long reaction time, while the total alcohol selectivity decreased from 17% to 13.5%. The selectivity of CO$_2$ and DME did not change much as the reaction proceeded. Figure 7c showed the product distribution on the CuFe/SiO$_2$-T catalyst. After 30 h of reaction, the selectivity of total alcohol was basically maintained at about 21%. This implies that the treatment of piranha solution not only contributed to the improvement of the alcohols' synthesis performance, but also endowed the catalyst with superior reaction stability. Figure 7d shows the STY of hydrocarbons on the two catalysts. The CuFe/SiO$_2$-N catalyst showed a sharp increase with time on stream, while the STY of hydrocarbons on the CuFe/SiO$_2$-T catalyst changed less. This also explained the gradual increase in CO conversion over the CuFe/SiO$_2$-N catalysts, as the increased CO conversion contributes to the hydrocarbon product.

The alcohol products' distributions among the total alcohols over the CuFe/SiO$_2$-N and CuFe/SiO$_2$-T catalysts are shown in Figure 7e,f, respectively. From Figure 7e, the selectivity of methanol on the CuFe/SiO$_2$-N catalyst gradually decreased with the increase in the reaction time, while the selectivity of ethanol and propanol gradually increased, with the selectivity of ethanol increasing from 22.4% to 29.9%. Due to the low content of C$_4$ and C$_{4+}$ alcohols, their selectivity did not change significantly during the whole reaction process. The variation of product selectivity indicates that the CuFe/SiO$_2$-N catalyst itself may undergo a dynamic structural evolution during the reaction. In contrast, as can be seen from Figure 7f, the ethanol selectivity on the CuFe/SiO$_2$-T catalyst was well maintained at ~24%, and the selectivity almost did not change when the reaction time reached as long as 96 h, indicating that the treatment with the piranha solution endowed the catalyst with a stable structure and good stability for the long-term CO hydrogenation reaction.

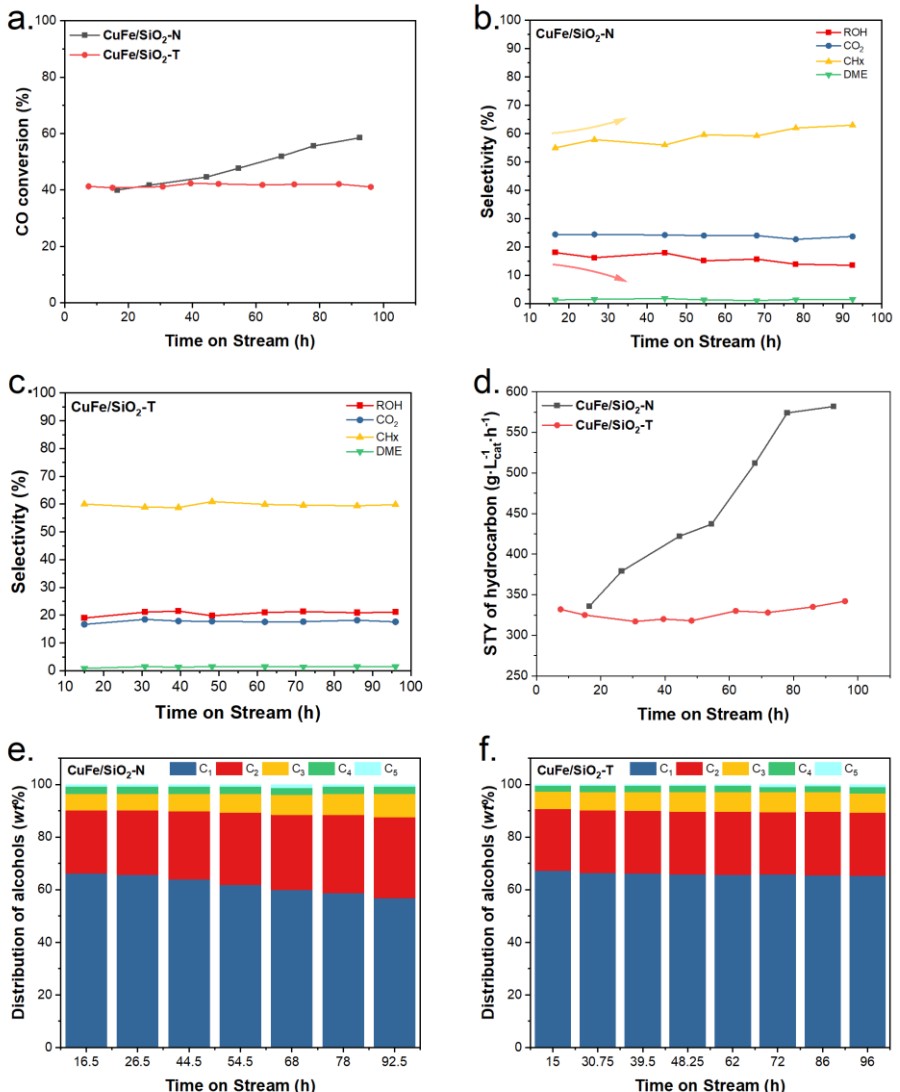

**Figure 7.** Catalytic performance of CuFe/SiO₂-N and CuFe/SiO₂-T catalysts: (**a**) CO conversion; (**b**,**c**) selectivity (C mol %) of different products; (**d**) STY of hydrocarbons; (**e**,**f**) and distribution of alcohols (250 °C, 5 MPa, 3000 h⁻¹).

## 3. Discussion

The characterization results implied that the treatment with piranha solution significantly affected the structure and surface properties of $SiO_2$. The specific surface area of the $SiO_2$-T catalyst increased significantly and the content of muti-coordinated hydroxyl groups on the surface also increased, which created advantages for the subsequent metal dispersion and anchoring on the surface of $SiO_2$. XRD, $H_2$-TPR, and STEM-EDS further revealed that after the treatment with piranha solution, the Cu and Fe active components were well dispersed in smaller particle sizes and firmly anchored on the surface of $SiO_2$ after the reduction and reaction process. It can be deduced that the muti-coordinated -OH might be the group that anchored the active component, and the increase of the hydroxyl group on the surface of $SiO_2$ was beneficial to the dispersion and anchoring of the active components Cu and Fe on the surface of $SiO_2$, thus the CuFe-$SiO_2$-T catalyst exhibited stable catalytic performance during the long-term reaction.

In addition, we also found that the treatment of piranha solution on $SiO_2$ affected the electronic state of Cu species on its surface. Combining the XRD, XPS, and in situ DRIFTS results, Cu existed in the form of $Cu^0$ and $Cu^+$. According to the mechanism of higher alcohol formation, a dual active site was required in the catalyst to form higher

alcohol synergistically, where one active site should provide non-dissociated CO for the CO insertion reaction and the other active center should provide the $CH_x$ species for the carbon chain growth [27]. As seen from the in situ DRIFTS results, the adsorption strength of CO on the CuFe/SiO$_2$-T catalyst was weaker, which avoided the cleavage of the C–O bond and excessive hydrogenation to form alkanes. Meanwhile, the adsorption peak of CO on $Cu^0$ could be obviously observed over the CuFe/SiO$_2$-T catalyst, indicating that more non-dissociated CO existed on the surface of the CuFe/SiO$_2$-T catalyst. The non-dissociative adsorption of CO was favorable to the CO insertion reaction, promoting the formation of alcohols. Hence, the total alcohol selectivity on the CuFe/SiO$_2$-T catalyst was found to be higher than that on the CuFe/SiO$_2$-N catalyst.

## 4. Materials and Methods

### 4.1. Materials

Cu(NO$_3$)$_2$·3H$_2$O and Fe(NO$_3$)$_3$·9H$_2$O were AR grade (analytical reagents) and used as the materials without further purification. SiO$_2$ (B-type silica gel) was purchased from Qingdao Haiyang Chemical Co., Ltd., China. Before use, the newly purchased SiO$_2$ was washed with deionized water several times, dried at 100 °C for 12 h, and calcined at 400 °C for 4 h. The piranha solution was prepared in the laboratory, composed of concentrated H$_2$SO$_4$ and 30% H$_2$O$_2$ solution (AR grade).

### 4.2. Preparation of the Catalysts

The piranha solution was prepared according to the literature method [49]. Specifically, 6 mL of 30% H$_2$O$_2$ solution was slowly dropped to 14 mL of concentrated H$_2$SO$_4$ under continuous stirring to obtain the mixture (7:3 volume ratio, concentrated H$_2$SO$_4$/30% H$_2$O$_2$) for use.

The piranha-solution-treated SiO$_2$ carrier (SiO$_2$-T) was prepared as follows: the SiO$_2$ was immersed for 30 min in the as-prepared piranha solution under a water bath temperature at 80 °C, then the SiO$_2$ was filtered and washed with distilled water until the pH = 7. The pretreated SiO$_2$ was dried at 100 °C for 12 h, and finally calcined at 400 °C for 4 h. The SiO$_2$ carrier without treatment by piranha solution was named as SiO$_2$-N, and the SiO$_2$ carrier after treatment was named as SiO$_2$-T.

The CuFe/SiO$_2$ catalyst (Cu/Fe molar ratio = 1:1) was prepared using an impregnation method. Typically, Cu(NO$_3$)$_2$·3H$_2$O and Fe(NO$_3$)$_3$·9H$_2$O were dissolved in deionized water to form a mixed solution. Then, the solution was poured into a SiO$_2$-N or SiO$_2$-T carrier (10 g) and sonicated for 1 h. The total moles of metal ions were maintained 0.03 mol per 10 g of SiO$_2$. The obtained CuFe/SiO$_2$ catalyst was dried at 100 °C for 12 h, and finally calcined at 400 °C for 4 h. The calcined powder was pelletized, crushed, and sieved into the size of 30–40 mesh. The Cu–Fe active components loading on the SiO$_2$-N and SiO$_2$-T carriers were named as CuFe/SiO$_2$-N and CuFe/SiO$_2$-T, respectively.

### 4.3. Characterization Techniques

The specific surface area (BET) of the catalyst was determined with a Micromeritics Tristar 3000 physisorption instrument, USA, (77 K N$_2$ adsorption) and was calculated from the desorption curve using the BET equation.

The X-ray diffraction (XRD) patterns were recorded on a Rigaku D/max-3B diffractometer (Bruker, Bremen, Germany) fitted with Cu K$\alpha$ radiation ($\lambda$ = 1.5404 Å).

The X-ray photoelectron spectra (XPS) were measured on a Vg Escalab MK-2 spectrometer (VG Scientific Ltd., UK) using Al K$\alpha$ rays (1486.6 eV) with an X-ray source accelerating at 12.5 kV and 250 W. The samples were pressed into 10 mm diameter, 1 mm-thick discs and fixed on a sample tray, and then the samples were transferred into a high vacuum chamber. The vacuum chamber was $2 \times 10^{-8}$ Pa and the binding energy of the contaminated carbon was C 1s = 284.6 eV.

The programmed temperature reduction (H$_2$-TPR) was carried out on the automatic chemical adsorption instrument (TP-5080, Tianjin Xianquan Corporation, China), and the

outgas was analyzed by a TCD detector. One hundred mg of the catalyst was heated to 300 °C in an Ar atmosphere during temperature-programmed heating and remained constant for 2 h. After cooling down to 50 °C, the reduction gas was switched to a mixture gas of 10 vol% $H_2$/Ar and programmed to 500 °C at a heating rate of 5 °C /min.

The surface -OH groups on the catalysts before and after treatment were investigated under Ar flow on a Bruker Tensor 27 FT-IR spectrometer (Bruker, Bremen, Germany) with a MCT detector (4000–600 $cm^{-1}$). Firstly, the sample (powder) was placed in an infrared cell equipped with a ZnSe window, and then the catalyst was treated at 300 °C in Ar flow for 1 h to remove the adsorbed water on the catalyst's surface. Finally, the catalyst was scanned to obtain the spectrum (KBr as reference).

In situ DRIFTS of CO adsorption was performed on the FT-IR spectrometer mentioned above. The catalyst placed in the chamber was firstly reduced by $H_2$ at 300 °C K for 2 h. Subsequently, the catalyst was flushed with Ar and cooled down to ambient temperature again, and then the background spectrum was collected. After the slow introduction of CO flow, the spectrum was obtained until it was unchanged with time. Then, CO was switched to Ar for purging the physically adsorbed species and the spectra were recorded with the purging time.

Scanning transmission electron microscopy (STEM) and energy-dispersive X-ray spectroscopy (EDS) measurements were performed on a Tecnai G2 F20 S-Twin electron microscope (FEI Company, Hillsboro, OR, USA) operated at an acceleration voltage of 200 kV. The sample was dispersed ultrasonically in ethanol for 5 min, and a drop of solution was deposited onto a molybdenum grid.

*4.4. Catalytic Performance*

The catalytic performance of the different catalysts for $C_{2+}$ alcohols' synthesis was evaluated in a fixed-bed stainless steel reactor. One mL of catalyst (30–40 mesh) was filled in and reduced using a mixture of 20 vol% $H_2$/Ar at 300 °C for over 4 h. After that, the temperature was cooled to 250 °C and feed gas $H_2$/CO (volume ratio of 2.5/1) was introduced under a gas hourly space velocity (GHSV) of 3000 $h^{-1}$. The data were collected every few hours and the products were separated by a cold trap. The liquid products were detected on two chromatographs equipped with a GDX-401 column (TCD) and a WondaCap WAX column (FID). The outlet gas was analyzed by a gas chromatograph equipped with a column of carbon sieves and a thermal conductivity detector (Shanghai Fanwei Instrument Equipment Co., Ltd., China) for $H_2$, $CH_4$, CO, and $CO_2$, and another chromatograph equipped with a column of GDX-403 and a flame ionization detector (Shanghai Precision Instrument Co., Ltd., Shanghai, China) for $CH_x$ mixtures ($C_1$, $C_2$, $C_3$, $C_4$, and $C_{4+}$), DME, and methanol, respectively.

**5. Conclusions**

In conclusion, the treatment of piranha solution on $SiO_2$ promoted a larger surface area and more muti-coordinated -OH groups on the surface, which was the anchor site for the active components, resulting in a better dispersion and firm combination of the active Cu and Fe components with the support. Therefore, the catalyst was endowed with good stability during the long-term reaction (~96 h). Moreover, in-situ DRIFTS further revealed that when the Cu component was loaded on the $SiO_2$ carrier treated with piranha solution, the CO adsorption on $Cu^+$ over the CuFe/$SiO_2$-T catalyst was weakened to avoid its excessive hydrogenation to form hydrocarbon products, while the non-dissociative adsorbed CO content on $Cu^0$ was significantly increased, which were both favorable to the formation of alcohol products. As a result, the total alcohol selectivity on the CuFe/$SiO_2$-T catalyst was higher than the CuFe/$SiO_2$-N catalyst. The preparation strategy reported here may bring new ideas to the design of supported catalysts with stabilized structures for a wide range of catalytic reactions.

**Author Contributions:** Conceptualization, N.G. and Y.W.; formal analysis, N.G. and Y.W.; writing—original draft preparation, N.G. and Y.W.; writing—review and editing, Y.W. and Y.T.; supervision, Y.T.; funding acquisition, Y.T. and Q.M. All authors have read and agreed to the published version of the manuscript.

**Funding:** This research was funded by the National Natural Science Foundation of China (22172182 and 21573269), the Natural Science Foundation of Shanxi Province (20210302123009), and the Foundation of State Key Laboratory of High Efficiency Utilization of Coal and Green Chemical Engineering (Grant No. 2022-K06).

**Acknowledgments:** The authors acknowledge the financial support of the National Natural Science Foundation of China (22172182 and 21573269), the Natural Science Foundation of Shanxi Province (20210302123009), and the Foundation of State Key Laboratory of High efficiency Utilization of Coal and Green Chemical Engineering (Grant No. 2022-K06).

**Conflicts of Interest:** The authors declare no conflict of interest.

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
