# Peer review of "A Simple Strategy Stabilizing for a CuFe/SiO2 Catalyst and Boosting Higher Alcohols’ Synthesis from Syngas"

_catalysts, doi:10.3390/catal13020237_

Round 1

Reviewer 1 Report

The main goal of the paper is to develop an effective strategy to enhance the stability of the CuFe/SiO2 catalyst and boost the synthesis of higher alcohols from syngas by pretreating SiO2 with piranha solution. The effect of the surface properties of SiO2 carrier on the dispersion of active components and synthesis performance of higher alcohols before and after the treatment were investigated.

This manuscript is well written. However, there are some gaps that need to be addressed before publication.

1. The research progress of materials for higher alcohols synthesis from syngas should be added in the Introduction. What is the main difference and innovation between the composite system and the literature? Some recent references should be cited, such as ACS Omega 2022, 7, 24, 21346–21356; Fuel Processing Technology 2015, 138, 305-313; Applied Catalysis B: Environmental 272, 2020, 118950

2. Which type of reactor was used? What was the residence time? The description of the methodology in Section should be clarified.

3. Surface composition (XPS) data should be supplemented with deconvolution.

4. The content of Fe and Cu should be add.

Reviewer 2 Report

The article is an experimental and theoretical study on catalytic conversion of syngas to alcohols. This is an interesting issue and worth of research work.

The authors make use of a set of well described characterization methods for the materials involved: raw materials, reaction products and catalysts. Results discussion and interpretation is quite satisfactory as well as the language of the article. Overall the work contributes in clarifying the reaction mechanism and the catalytic action of the Cu and Fe metal catalysts.

One point of concern is that authors do not present any information about the piranha used in solution for impregnating the SiO2. Since this material plays an important role in the catalytic action, it must be characterized in order to justify its use and discuss the surface modifications that were found after impregnation as mentioned in part 3.

Some other comments:

An interesting result is on Figure 7a and it is the increasing trend of CO. It is worth of commenting this trend.

Definition of selectivities, specifying if they are given in mol or mass fractions, should be useful.

Also specify if gas compositions such as H2/CO are in mass or mol fractions.

Page 8, line 254: 0,03 mol, is per g of solid?

Reviewer 3 Report

Comments to the Authors

In this manuscript authors reported an effective strategy to enhance the stability of the CuFe/SiO2 catalyst and boost the synthesis of higher alcohols from syngas by pretreating SiO2 with piranha solution. This research has value for the researchers in the related areas. However, the paper needs improvement before acceptance for publication. My detailed comments are as follow:

1.      In the introduction section authors should introduced following relevant catalytic related articles using SiO2 and iron :

a.       doi.org/10.1016/j.jece.2018.10.067

b.      doi.org/10.1016/j.colcom.2019.100218

2.      In XRD authors should mention the JCPDS no of materials.

3.      In the histograms of recycling studies authors should provide error bar.

4.      There are few typos and grammatical errors.

5.      The writing of conclusion section should be improved.

Round 2

Reviewer 2 Report

Please give any information available on piranha material:

where it comes from?

its composition

its physical form

any other info available

Author Response

Response to Reviewer 2 Comments

Point 1. Please give any information available on piranha material:

where it comes from? Its composition, its physical form, any other info available

Response 1: Thank you for your suggestion. We feel sorry that we did not express the preparation of piranha material clearly.

In the experiment part, we divided the “4.1. Preparation of the CuFe/SiO2 catalysts” part into two parts: “4.1. Materials” and “4.2. Preparation of the catalysts” to make it clearer (marked blue on page 9 line 288 and 295). We also additionally mentioned piranha solution was prepared in the laboratory and the composition of piranha solution was concentrated H2SO4 and 30% H2O2 solution (AR grade) (marked blue on page 9 line 292-294), and we further supplemented the detailed preparation process of piranha solution and cited the reference literature to the preparation part (marked blue on page 9 line 296-299).

It should also be noted that “T-SiO2” was obtained by immersing SiO2 in the as-prepared piranha solution, please see more details in our revised manuscript (marked blue on page 9 line 301).

Besides, as for the structural information such as physical form of the piranha solution treated materials (T-SiO2 and CuFe/ T-SiO2), BET and FTIR spectroscopy measurements were first applied to investigate the changes of textual and surface properties (marked blue on page 2 line 78 and page 3 line 84), resulting in a larger surface area and more muti-coordinated -OH groups on SiO2 surface after the treatment of piranha solution. Furthermore, the SiO2-supported CuFe/SiO2 catalysts were systematically characterized by BET, XRD, STEM-EDS, XPS, H2-TPR to reveal the effect of piranha solution treated SiO2 on the dispersion and anchoring of active components. We appreciate the Reviewer’s warm work earnestly and hope that the revision will meet with approval.

Thank you again for your nice and valuable comments.
